# A randomised double blind placebo controlled phase 2 trial of adjunctive aspirin for tuberculous meningitis in HIV-uninfected adults

Nguyen TH Mai[1,2], Nicholas Dobbs[3], Nguyen Hoan Phu[1,2], Romain A Colas[4], Le TP Thao[1], Nguyen TT Thuong[1], Ho DT Nghia[1,2], Nguyen HH Hanh[1,2], Nguyen T Hang[1], A Dorothee Heemskerk[1,5], Jeremy N Day[1,6], Lucy Ly[4], Do DA Thu[1], Laura Merson[6], Evelyne Kestelyn[1,6], Marcel Wolbers[1], Ronald Geskus[1,6], David Summers[3], Nguyen VV Chau[1,2], Jesmond Dalli[4], Guy E Thwaites[1,6]*

[1]Oxford University Clinical Research Unit, Ho Chi Minh City, Vietnam; [2]Hospital for Tropical Diseases, Ho Chi Minh City, Vietnam; [3]Western General Hospital, Edinburgh, United Kingdom; [4]Lipid Mediator Unit, William Harvey Research Institute, Barts and The London School of Medicine and Dentistry, Queen Mary University of London, London, United Kingdom; [5]Department of Medical Microbiology and Infection Control, VU medical centre, VU University Amsterdam, Amsterdam, Netherlands; [6]Centre for Tropical Medicine and Global Health, Nuffield Department of Medicine, University of Oxford, Oxford, United Kingdom

**Abstract** Adjunctive dexamethasone reduces mortality from tuberculous meningitis (TBM) but not disability, which is associated with brain infarction. We hypothesised that aspirin prevents TBM-related brain infarction through its anti-thrombotic, anti-inflammatory, and pro-resolution properties. We conducted a randomised controlled trial in HIV-uninfected adults with TBM of daily aspirin 81 mg or 1000 mg, or placebo, added to the first 60 days of anti-tuberculosis drugs and dexamethasone (NCT02237365). The primary safety endpoint was gastro-intestinal or cerebral bleeding by 60 days; the primary efficacy endpoint was new brain infarction confirmed by magnetic resonance imaging or death by 60 days. Secondary endpoints included 8-month survival and neuro-disability; the number of grade 3 and 4 and serious adverse events; and cerebrospinal fluid (CSF) inflammatory lipid mediator profiles. 41 participants were randomised to placebo, 39 to aspirin 81 mg/day, and 40 to aspirin 1000 mg/day between October 2014 and May 2016. TBM was proven microbiologically in 92/120 (76.7%) and baseline brain imaging revealed $\geq$1 infarct in 40/114 (35.1%) participants. The primary safety outcome occurred in 5/36 (13.9%) given placebo, and in 8/35 (22.9%) and 8/40 (20.0%) given 81 mg and 1000 mg aspirin, respectively (p=0.59). The primary efficacy outcome occurred in 11/38 (28.9%) given placebo, 8/36 (22.2%) given aspirin 81 mg, and 6/38 (15.8%) given 1000 mg aspirin (p=0.40). Planned subgroup analysis showed a significant interaction between aspirin treatment effect and diagnostic category ($P_{heterogeneity}$ = 0.01) and suggested a potential reduction in new infarcts and deaths by day 60 in the aspirin treated participants with microbiologically confirmed TBM (11/32 (34.4%) events in placebo vs. 4/27 (14.8%) in aspirin 81 mg vs. 3/28 (10.7%) in aspirin 1000 mg; p=0.06). CSF analysis demonstrated aspirin dose-dependent inhibition of thromboxane $A_2$ and upregulation of pro-resolving CSF protectins. The addition of aspirin to dexamethasone may improve outcomes from TBM and warrants investigation in a large phase 3 trial.

*For correspondence:
gthwaites@oucru.org

Competing interests: The authors declare that no competing interests exist.

**eLife digest** The deadliest form of tuberculosis is tuberculosis meningitis (TBM), which causes inflammation in the brain. Even with the best treatment available, about half of patients with TBM become disabled or die, often because they have a stroke. Strokes are caused by blood clots or other blockages in blood vessels in the brain. Aspirin is known to prevent blood clots and helps reduce inflammation. Some scientists wonder if it might help patients with TBM by preventing blockages in blood vessels.

Now, Nguyen et al. show that adding aspirin to existing TBM treatments may reduce strokes in some patients. In the experiments, 120 patients with TBM were randomly assigned to receive a low dose of aspirin (81 mg/day), a high dose of aspirin (1000mg/day), or an identical tablet that contained no medication. All the patients also took the anti-tuberculosis drugs and steroids usually used to treat the condition. Both doses of aspirin appeared to be safe. Patients who received aspirin were less likely to have a stroke or die in the first two months of treatment than patients who received the fake pill. But the difference was so small it could have been caused by chance.

In the 92 patients with clear evidence of tuberculosis bacteria in their brains, the benefit of aspirin was larger and unlikely to be due to chance. The benefit was greatest for those who received the higher dose of aspirin, only 10.7% of these patients died or had a stroke, compared with 14.8% of those who received a low dose of aspirin, or 34% of those who received the fake pill. Next, Nguyen et al. looked at brain fluid taken from the TBM patients before and after they received the aspirin or fake medication. The experiments showed that patients treated with high dose aspirin had much lower levels of a clot-promoting substance called thromboxane A2 and more anti-inflammatory molecules.

Larger studies are needed in children and adults to confirm that aspirin helps prevent strokes or death in patients with TBM. Studies are also needed on patients who have both TBM and HIV infections. But if more studies show aspirin is safe and effective, adding this medication to TBM treatment may be an inexpensive way to prevent death or disability.

## Introduction

New host-directed therapies are urgently required for all forms of tuberculosis, but especially for tuberculous meningitis (TBM), the most lethal form of the disease, which kills or disables around half of sufferers (*Thwaites, 2013*). Therapeutic strategies to improve outcomes can be broadly divided into those directed against the bacteria and their enhanced killing, and those directed at the host and the control of the inflammatory response. To date, attempts to optimise bacteria-directed anti-tuberculosis regimens have not been shown to clearly benefit patients with TBM (*Ruslami et al., 2013*; *Heemskerk et al., 2016*). In contrast, host-directed therapy with adjunctive anti-inflammatory corticosteroids has been shown to reduce mortality from TBM, although without reduced disability amongst survivors (*Prasad et al., 2016*). There is, therefore, an urgent need to explore alternative therapeutic strategies that may prevent the long-term neurological sequelae of TBM and complement the short-term survival benefits of dexamethasone.

Cerebral infarction is the commonest cause of irreversible neurological injury from TBM (*Lammie et al., 2009*). TBM-related infarcts are typically located in the territories of the proximal middle cerebral artery and the medial lenticulostriate and thalamoperforating vessels, where the basal meningeal inflammatory exudate is most intense (*Lammie et al., 2009*; *Hektoen, 1896*; *Misra et al., 2011*). Their pathogenesis remains controversial, in particular the role of vessel thrombosis. Some autopsy studies have either failed to find arterial thrombosis associated with infarcts, or found it to be uncommon (*Doniach, 1949*); whereas others have reported that thrombosis is common, especially when associated with tuberculous vasculitis (*Poltera, 1977*). The limited available evidence suggests that TBM-related infarcts are caused by a combination of vasospasm, intimal proliferation, and thrombosis (*Lammie et al., 2009*).

Aspirin acts by irreversibly inhibiting the cyclooxygenase pathway of arachidonic acid metabolism and the production of prostanoids (*Vane, 1971*). Low dose aspirin (75–150 mg) prevents ischaemic

cerebrovascular disease (*Richman and Owens, 2017*) and higher dose aspirin (up to four grams daily) is used for the treatment of some inflammatory conditions (e.g. rheumatic fever) (*Cilliers et al., 2015*). Its anti-inflammatory effects are thought to occur at doses >600 mg daily, through the inhibition of pro-inflammatory prostaglandins (e.g. $PGE_2$, $PGF_{2\alpha}$ and $PGD_2$) and the unstable prostanoid, thromboxane $A_2$ ($TXA_2$) (*Botting, 2010*). Low dose aspirin causes less inhibition of pro-inflammatory prostaglandins, but causes clinically significant inhibition of $TXA_2$ and platelet aggregation. Until recently, the inhibitory effect on platelets and thrombus formation was thought to explain aspirin's well-documented reduction in the risk of death from stroke and myocardial infarction (*Raju et al., 2011*). However, these effects may be augmented by aspirin's ability to trigger the production of 15-epi-lipoxins, 17R-resolvins and protectins, molecules that alongside the recently discovered maresins actively promote the resolution of inflammation (*Spite and Serhan, 2010*). The 'pro-resolution' properties of aspirin are not shared with any other non-steroidal anti-inflammatory drugs (NSAID), or corticosteroids. It represents a potentially unique mode of action by which aspirin, alongside the prevention of thrombosis, might prevent infarctions and speed resolution of intracerebral inflammation and improve outcomes from TBM. Furthermore, there are intriguing data from murine models of tuberculosis which suggest aspirin and other non-steroidal anti-inflammatory drugs may enhance *Mycobacterium tuberculosis* killing (*Byrne et al., 2007*; *Vilaplana et al., 2013*).

Two previous trials of adjunctive aspirin for TBM have been reported. The first randomised 118 Indian adults with TBM to standard anti-tuberculosis chemotherapy, with or without aspirin (150 mg daily) (*Misra et al., 2010*). By 3 months, brain magnetic resonance imaging (MRI) proven infarction occurred in 13 (43%) in the placebo arm and 8 (24%) in the aspirin group (p=0.18). Aspirin was associated with a reduction in mortality (43% versus 22%, p=0.02) without a significant increase in adverse events. The results are hard to interpret, however, because of the variable use of prednisolone between the treatment arms. Participants who received prednisolone and aspirin appeared to benefit the most. The second trial randomised 146 South African children with TBM to standard anti-tuberculosis chemotherapy plus placebo (n = 50), low-dose aspirin (75 mg/day) (n = 47), or high-dose aspirin (100 mg/kg/day) (n = 49) (*Schoeman et al., 2011*). Aspirin had no significant impact on survival, motor or cognitive outcomes.

We hypothesised that aspirin prevents TBM-related brain infarction by its anti-thrombotic, anti-inflammatory, and pro-resolution effects. We chose to investigate two aspirin doses: a low dose (81 mg/day) with anti-thrombotic but minimal anti-inflammatory activity; and a higher dose (1000 mg/day) with both anti-thrombotic and anti-inflammatory activity. Our primary objective was to demonstrate the safety, tolerability, and potential efficacy of 81 mg and 1000 mg aspirin when added to dexamethasone for the first 60 days of TBM treatment. Our secondary objective was to investigate the potential mechanisms of actions of the two aspirin doses by examining the profiles of lipid mediators, including the pro-inflammatory eicosanoids and aspirin triggered pro-resolving mediators, in the cerebrospinal fluid (CSF) of participants.

## Results

Between October 2014 and May 2016, 192 patients were assessed for eligibility and 120 were randomised (*Figure 1* and *supplementary file 2*): 41 received placebo, 39 received aspirin 81 mg/day, and 40 received aspirin 1000 mg/day. Two participants in the placebo group were lost to follow-up after 57 and 217 days, respectively. Gastro-intestinal bleeding event data were missing for seven participants (three placebo, 4 aspirin 81 mg) because they either died or were lost to follow-up before day 60. MRI-proven new intracranial bleeding event or infarct data were missing for 15 subjects (six placebo, 7 Aspirin 81 mg, 2 Aspirin 1000 mg) because they either died (four placebo, 6 aspirin 81 mg, 1 aspirin 1000 mg), were lost to follow-up before day 60 (1 placebo), or they were too unwell to have scans within 60 ± 10 days (two placebo, 1 aspirin 81 mg, 2 aspirin 1000 mg).

The per-protocol population excluded 22 participants (*Figure 1*): two had a confirmed alternative diagnosis (*Listeria monocytogenes* meningitis and *Mycobacterium avium intracellulare* meningitis), one had confirmed MDR TBM, and 19 received <30 days of study drug for reasons other than death (five placebo; 8 aspirin 81 mg; 9 aspirin 1000 mg).

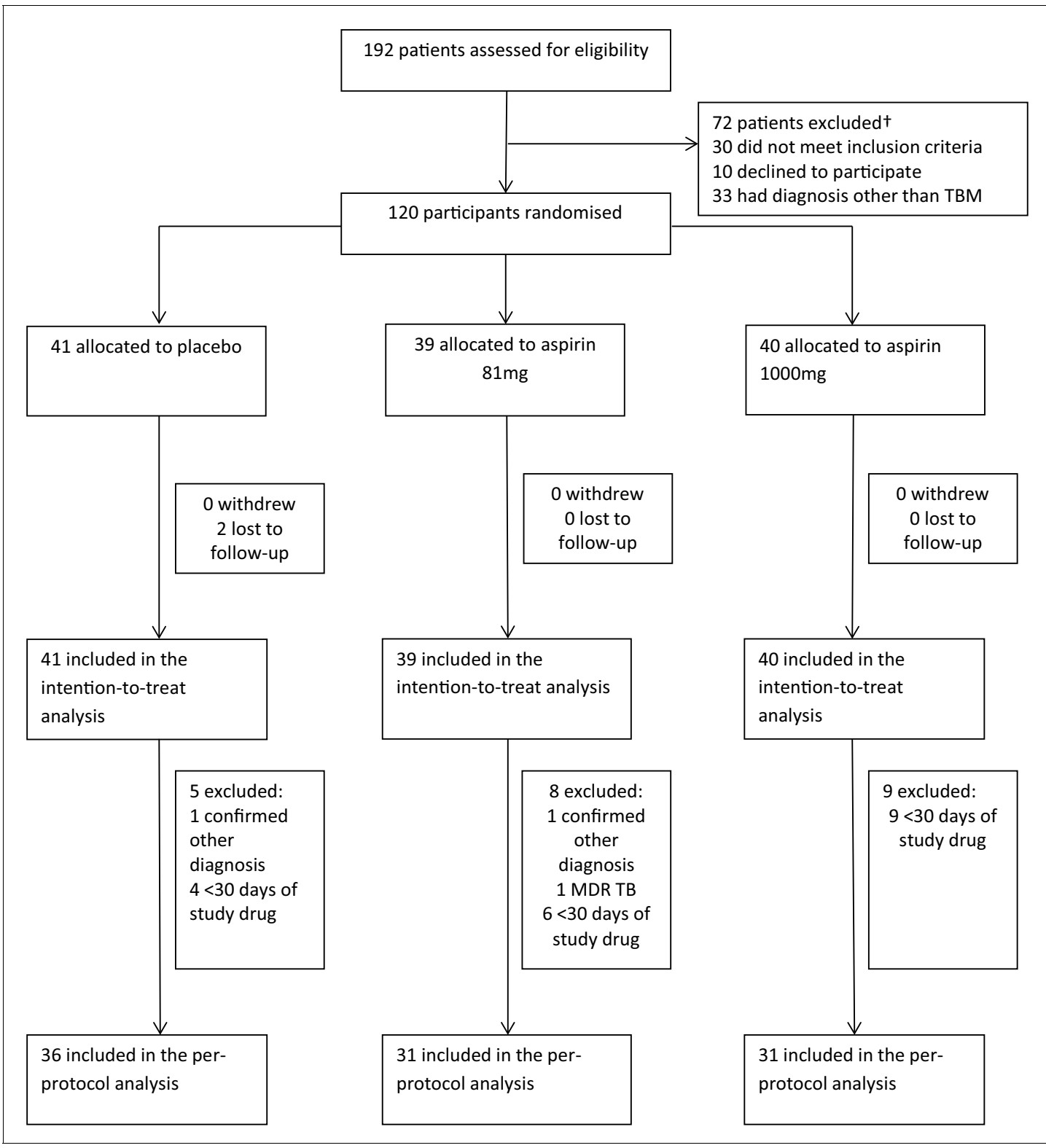

**Figure 1.** Participant flow through the trial.

## Baseline characteristics

Baseline characteristics were balanced between the three treatment groups (*Table 1*). The predominant gender was male (65.8%), the median age was 41 years and duration of illness was 10 days. Most participants had mild to moderate illness severity, with only 15 (12.5%) MRC grade III at enrollment. Baseline MRI revealed ≥1 infarct in 40/114 (35.1%) participants; the placebo and aspirin 1000 mg groups had a higher proportion with infarcts (42.5% and 38.5% respectively) than the aspirin 81 mg group (22.9%). According to the published diagnostic criteria 92/120 (76.7%) had definite TBM, 17/120 (14.2%) probable, and 9/120 (7.5%) possible TBM (*Marais et al., 2010*). Of the 92 participants with definite TBM, 42 (45.7%) had *M. tuberculosis* cultured from the CSF and acid-fast bacilli were seen in the CSF in 50 others (54.3%). Amongst patients with culture-confirmed disease, 10/42 (23.8%) had isoniazid resistant infection and one (2.4%) had MDR TBM.

## Primary outcomes

The primary safety outcome of gastro-intestinal or cerebral bleeding occurred in 21/111 (18.9%): 5/36 (13.9%) given placebo, and in 8/35 (22.9%) and 8/40 (20.0%) given 81 mg and 1000 mg aspirin respectively (p=0.59) (*Table 2*). Only one new cerebral bleed occurred (in the aspirin 81 mg group): an asymptomatic micro-haemorrhagic transformation of a lacunar infarct. Four gastro-intestinal bleeding events were defined as either serious, or grade 3 or 4: two serious haematemesis events in the placebo group and one in the aspirin 81 mg group, and one grade 3 episode of melena in the aspirin 1000 mg. The majority of bleeding events (16/20; 80%) were defined as either grade 1 or two events, 15/16 described as >5 mls of changed or fresh blood aspirated from a nasogastric tube, and 1 episode of melena (*supplementary file 3*). None of these events required active management, but on each occasion the study drug was stopped immediately.

The primary efficacy outcome of new MRI-proven brain infarction or death occurred in 25/112 (22.3%): 11/38 (28.9%) given placebo, 8/36 (22.2%) given aspirin 81 mg, and 6/38 (15.8%) given 1000 mg aspirin. The observed absolute risk reductions in the aspirin 81 mg and aspirin 1000 mg groups versus placebo were −6.7% (95% confidence interval (CI) −25.7% to +13.1%) and −13.2% (95% CI −31.0% to 5.7%), respectively, although the differences were not statistically significant (p=0.40) (*Table 2*).

The observed risk of a new MRI-proven brain infarction was lower in the aspirin treated patients compared to placebo, although not statistically significant (p=0.18) (*Table 2*). In addition, 9/15 (60.0%) of brain infarcts seen at baseline in the aspirin 1000 mg group resolved by day 60, whereas resolution only occurred in 1/7 (14.2%) in the aspirin 81 mg group and 6/14 (42.9%) in the placebo group (p=0.14). There was only one death in the aspirin 1000 mg treated participants by day 60.

In the per-protocol population new infarction or death occurred by day 60 in 19/95 (20.0%): 10/34 (29.4%) given placebo, 6/31 (19.3%) given aspirin 81 mg, and 3/30 (10.0%) given 1000 mg aspirin (p=0.16) (*Table 3*). The observed absolute risk reductions in the aspirin 81 mg and aspirin 1000 mg groups versus placebo were −10.1% (95% CI −29.7% to +11.0%) and −19.4% (95% CI −37.4% to +0.6%), respectively (p=0.16). No deaths occurred in the aspirin 1000 mg group, compared to 13% and 11% in the aspirin 81 mg and placebo groups respectively (p=0.11).

## Planned sub-group analyses

Planned sub-group analyses for the primary efficacy outcome are reported in *Table 4*. No clear subgroup signal was seen for any subgrouping variable except for the diagnostic category which showed a significant interaction with the aspirin treatment effect (P$_{heterogeneity}$ = 0.01) and suggested a potential reduction in new infarcts and deaths by day 60 in the aspirin-treated participants with microbiologically confirmed TBM (11/32 (34.4%) events in placebo vs. 4/27 (14.8%) in aspirin 81 mg vs. 3/28 1000 (10.7%) in aspirin 1000 mg; p=0.06). These beneficial effects were most marked in the aspirin 1000 mg group (aspirin 81 mg vs placebo: odds ratio (OR) 0.33, 95% CI 0.09–1.20, p=0.093; aspirin 1000 mg vs. placebo: OR 0.23, 95% CI 0.06–0.93, p=0.039) (*Figure 2*). These effects equate to the number-needed-to-treat (NNT) to prevent an infarct or death by day 60 of 5 for aspirin 81 mg and 4 for aspirin 1000 mg.

**Table 1.** Baseline characteristics.

| Variable | Placebo (n = 41) | 81 mg Aspirin (n = 39) | 1000 mg aspirin (n = 40) |
|---|---|---|---|
| Age (years) – median(IQR) | 43 (33–49) | 39 (34–48) | 40 (31–53) |
| Male gender- no.(%) | 27 (65.9) | 28 (71.8) | 24 (60.0) |
| Weight (kg) | 50 (44–60) | 50 (45–60) | 50 (47–58) |
| Previous tuberculosis treatment- no.(%) | 3 (7.3) | 1 (2.6) | 1 (2.5) |
| Illness duration (days) – median(IQR) | 10 (9–14) | 10 (8–15) | 10 (8–14) |
| MRC Grade: – no.(%)* | | | |
| - I | 16 (39.0) | 15 (38.5) | 15 (37.5) |
| - II | 20 (48.8) | 19 (48.7) | 20 (50.0) |
| - III | 5 (12.2) | 5 (12.8) | 5 (12.5) |
| Glasgow Coma Score (/15) – median(IQR) | 15 (14–15) | 15 (14–15) | 15 (14–15) |
| Cranial nerve palsy- no.(%) | 12 (29.3) | 9 (23.1) | 12 (30.0) |
| Hemiplegia- no.(%) | 3 (7.3) | 0 | 1 (2.5) |
| Paraplegia- no.(%) | 1 (2.4) | 4 (10.3) | 1 (2.5) |
| Chest radiograph: - no.(%) | | | |
| - Normal | 20 (48.8) | 25 (64.1) | 19 (47.5) |
| - Miliary tuberculosis | 10 (25.0) | 3 (7.9) | 11 (28.2) |
| - Other lung tuberculosis | 11 (26.8) | 10 (25.6) | 9 (22.5) |
| Plasma sodium (mmol/L) – median(IQR) | 127 (124–131) | 130 (125–134) | 129 (125–132) |
| CSF: – median(IQR) | | | |
| Total leucocyte count (/mm$^3$) | 311 (126–425) | 328 (120–605) | 180 (141–340) |
| % neutrophils | 22 (7–49) | 12 (6–29) | 14 (6–32) |
| % lymphocytes | 78 (51–93) | 88 (71–94) | 84 (68–94) |
| Total protein (g/dL) | 1.4 (1.1–2.0) | 1.2 (0.9–1.9) | 1.6 (1.1–2.1) |
| Lactate | 5.0 (4.0–6.6) | 5.0 (3.5–6.9) | 4.9 (3.5–6.0) |
| Glucose | 2.1 (1.3–2.8) | 2.2 (1.7–2.9) | 2.4 (1.7–2.8) |
| CSF:plasma glucose | 0.3 (0.2–0.4) | 0.3 (0.2–0.5) | 0.4 (0.3–0.5) |
| Diagnostic category: - no.(%)[†] | | | |
| - Definite | 34 (82.9) | 29 (74.4) | 29 (72.5) |
| - Probable | 4 (9.8) | 5 (12.8) | 8 (20.0) |
| - Possible | 2 (4.9) | 4 (10.3) | 3 (7.5) |
| - Confirmed other diagnosis | 1 (2.4) | 1 (2.6) | 0 |
| Brain imaging performed: - no.(%) | 40 | 35 | 39 |
| - Normal | 15 (37.5%) | 13 (37.1%) | 17 (43.6%) |
| - Meningeal enhancement | 6 (15.0%) | 5 (14.3%) | 3 (7.7%) |
| - Tuberculomas | 7 (17.5%) | 4 (11.4%) | 4 (10.3%) |
| - Hydrocephalus | 6 (15.0%) | 4 (11.4%) | 3 (7.7%) |
| - Infarcts | 17 (42.5%) | 8 (22.9%) | 15 (38.5%) |
| DST available– no.[‡] | 19 | 9 | 14 |
| - No isoniazid or rifampicin resistance – no.(%)[§] | 12 (63.2) | 7 (77.8) | 12 (86.7) |
| - Isoniazid resistant- no.(%) | 6 (31.6) | 1 (11.1) | 2 (14.3) |
| - Rifampicin resistant- no.(%) | 1 (5.2) | 0 | 0 |
| - MDR- no.(%) | 0 | 1 (11.1) | 0 |
| Initial anti-tuberculosis drug treatment- no.(%)[#] | | | |
| - RHZES | 40 (97.6) | 37 (94.9) | 40 (100.0) |
| - RHZE | 1 (2.4) | 1 (2.6) | |
| - RHZL | 0 | 1 (2.6) | |
| LTA4H genotype available – no | 41 | 38 | 37 |
| - CC | 21 (51.2) | 14 (36.8) | 16 (43.2) |
| - CT | 16 (39.0) | 18 (47.4) | 20 (54.1) |
| - TT | 4 (9.8) | 6 (15.8) | 1 (2.7) |

IQR = inter quartile range

*MRC denotes modified British Medical Research Council criteria. Grade I indicates a Glasgow coma score of 15 with no neurologic signs, grade II a score of 11 to 14 (or 15 with focal neurologic signs), and grade III a score of 10 or less.

[†]Diagnostic categories were assigned according to the consensus case definition (table S1) (**Marais et al., 2010**). Confirmed other diagnosis was only made based on microbiological evidence.

[‡]DST = drug susceptibility test.

§MDR (multidrug-resistance) is defined as resistance to at least both isoniazid and rifampicin. In all categories, other resistance may be present.

#Rifampicin (R), Isoniazid (H), Pyrazinamide, Ethambutol (E), Streptomycin (S).

## Secondary outcomes and adverse events

In the ITT population there was no significant difference in death or disability by day 60 or month eight across the treatment groups (*Table 5*). The 8-month mortality was 14/118 (11.9%): 5/39 (12.8%) in the placebo group versus 6/39 (15.4%) and 3/40 (7.5%) in the 81 mg and 1000 mg aspirin groups respectively (p=0.50; *Figure 2* panels). Although the observed mortality was lowest in the aspirin 1000 mg group at 60 days and 8 months, the proportion of participants in this group with moderate (6/40 (15.0%)) or severe disability (2/40 (5.0%)) by 8 months was not significantly different from the aspirin 81 mg (4/39 (10.3%) and 2/39 (5.1%)) and placebo treated participants (7/38 (18.4%) and 0/38 (0.0%)) (*supplementary file 4*).

In the per-protocol population, however, there was a trend to better 8 month outcomes in the aspirin 1000 mg group (p=0.13)(*Table 5*). Aspirin at either dose was not associated with a significant reduction in hospital stay (median 32 days for each group; p=0.84).

**Table 2.** Primary safety and efficacy outcomes by 60 days from randomisation in the intention-to-treat population.

| | Placebo (n = 41) | Aspirin 81 mg (n = 39) | Aspirin 1000 mg (n = 40) | Absolute risk difference [%] (95% confidence interval) | Overall comparison P-value |
|---|---|---|---|---|---|
| Primary safety outcomes | | | | | |
| Gastro-intestinal bleeding or MRI-proven intracranial bleeding event* | 5/36 (13.9%) | 8/35 (22.9%) | 8/40 (20.0%) | Aspirin 81 mg vs placebo: 9.0% (-9.3 to 26.9%) Aspirin 1000 mg vs placebo: 6.1% (-11.5 to 22.8%) | 0.59 |
| Gastro-intestinal bleeding event | 5/38 (13.2%) | 7/35 (20.0 %) | 8/40 (20.0 %) | Aspirin 81 mg vs placebo: 6.8% (-10.5 to 24.4%) Aspirin 1000 mg vs placebo: 6.8% (-10.2 to 23.4%) | 0.71 |
| MRI-proven intracranial bleeding event | 0/35 (0%) | 1/32 (3.1%) | 0/38 (0%) | Aspirin 81 mg vs placebo: 3.1% (-7.1 to 15.7%) Aspirin 1000 mg vs placebo: 0.0% (-9.9 to 9.2%) | 0.30 |
| Primary efficacy outcomes | | | | | |
| New MRI-proven brain infarction or death | 11/38 (28.9%) | 8/36 (22.2%) | 6/38 (15.8%) | Aspirin 81 mg vs placebo: −6.7% (-25.7 to 13.1%) Aspirin 1000 mg vs placebo: −13.2% (-31.0 to 5.7%) | 0.40 |
| New MRI-proven brain infarction† | 8/35 (22.9%) | 2/30 (6.7%) | 5/37 (13.5%) | Aspirin 81 mg vs placebo: −16.2% (-33.1 to 2.0%) Aspirin 1000 mg vs placebo: −9.3% (-27.2 to 8.7%) | 0.18 |
| Death | 4/41 (9.8%) | 6/39 (15.4%) | 1/40 (2.5%) | Aspirin 81 mg vs placebo: 5.6% (-9.5 to 21.1%) Aspirin 1000 mg vs placebo: −7.3% (-20.2 to 4.7%) | 0.14 |

*Gastro-intestinal bleeding event data are missing for seven participants (three placebo, 4 aspirin 81 mg) because they either died or were lost to follow-up before day 60. MRI-proven new intracranial bleeding event or infarct data are missing for 15 subjects (6 Placebo, 7 Aspirin 81 mg, 2 Aspirin 1000 mg) because they either died (four placebo, 6 aspirin 81 mg, 1 aspirin 1000 mg), were lost to follow-up before day 60 (1 placebo), or they were too unwell to have scans within 60 ± 10 days (two placebo, 1 aspirin 81 mg, 2 aspirin 1000 mg). A participant was excluded from the analysis of the combined primary safety endpoint if they had missing data for both gastro-intestinal bleeding event and MRI-proven intracranial bleeding event, or if information about one event type is missing and the other event type did not occur (i.e. participants for which it is unclear due to missing data whether the combined event occurred or not are excluded to avoid under-estimation of the true safety risk).

†18* 18 participants (6 Placebo, 9 Aspirin 81 mg, 3 Aspirin 1000 mg) did not have MRI information at either baseline (±7 days) or day 60 (±10 days). For death status, the patient lost to follow-up after 57 days was treated as being alive. Patients were excluded from the combined primary efficacy endpoint if they were alive but MRI information was missing. One patient (Placebo) had both a new MRI-proven brain infarction event and death.

**Table 3.** Primary efficacy outcomes by 60 days from randomisation in the per-protocol population

|  | Placebo (n = 36) | Aspirin81mg (n = 31) | Aspirin1000mg (n = 31) | Absolute risk difference [%] (95% confidence interval) | Overall comparison P-value |
|---|---|---|---|---|---|
| New MRI-proven brain infarction or death* | 10/34 (29.4%) | 6/31 (19.4%) | 3/30 (10.0%) | Aspirin 81 mg vs placebo: −10.1% (-29.7 to 11.0%) Aspirin 1000 mg vs placebo: −19.4% (-37.4 to 0.6%) | 0.16 |
| New MRI-proven brain infarction | 7/31 (22.6%) | 2/27 (7.4%) | 3/30 (10.0%) | Aspirin 81 mg vs placebo: −15.2% (-33.2 to 4.3%) Aspirin 1000 mg vs placebo: −12.6% (-31.0 to 6.6%) | 0.22 |
| Death | 4/36 (11.1%) | 4/31 (12.9%) | 0/31 (0%) | Aspirin 81 mg vs placebo: 1.8% (-14.4 to 19.1%) Aspirin 1000 mg vs placebo: −11.1% (-25.3 to 1.8%) | 0.11 |

*10 participants (five placebo, 4 aspirin 81 mg, 1 aspirin 1000 mg) did not have MRI information at either baseline (±7 days) or day 60 (±10 days). One participant (placebo) had both a new MRI-proven brain infraction event and death.

MRI brain imaging abnormalities (other than infarcts) were similar between the groups by day 60 and 8 months (*supplementary file 5*). Hydrocephalus, however, was less common by day 60 in the aspirin 1000 mg group (2/38 (5.3%)) than the aspirin 81 mg (5/32 (15.6%)) and placebo groups (8/35 (22.9%)(p=0.09). None of the participants in the trial with hydrocephalus underwent ventriculoperitoneal shunting. The proportion of participants in each group with infarcts by month eight did not differ significantly.

Overall, aspirin was not associated with a significant increase in grade 3 or four or serious adverse events, with the possible exception of more cardiac events in the aspirin groups (p=0.08) (*Table 6*). The numbers of participants with ≥1 serious adverse event were 12 (29.3%) in the placebo arm, 15 (38.5%) in the aspirin 81 mg arm and 9 (22.5%) in the 1000 mg arm (p=0.31). Adverse events resulting in study drug stop or interruption occurred in 7 (17.1%) given placebo, 10 (25.6%) 81 mg aspirin, and 10 (25.0%) 1000 mg aspirin (p=0.56). The commonest reason was mild gastro-intestinal bleeding (20/27; *supplementary file 3*). Hyponatraemia (plasma sodium <125 mmol/L) was more common in

**Table 4.** Sub-group analyses of the primary efficacy outcome in the intention-to-treat population.

|  | Placebo (events/n (risk%)) | Aspirin 81 mg (events/n (risk%)) | Aspirin 1000 mg (events/n (risk%)) | P-value comparison | P-heterogeneity |
|---|---|---|---|---|---|
| Diagnostic criteria |  |  |  |  |  |
| - Definite | 11/32 (34.4%) | 4/27 (14.8%) | 3/28 (10.7%) | 0.06 | 0.01 |
| - Probable/Possible | 0/6 (0%) | 3/8 (37.5%) | 3/10 (30.0) | 0.30 |  |
| MRC Grade |  |  |  |  |  |
| - I | 4/16 (25.0%) | 1/15 (6.7%) | 1/14 (7.1%) | 0.33 | 0.44 |
| - II | 4/17 (23.5%) | 4/17 (23.5%) | 4/19 (21.1%) | 1.00 |  |
| - III | 3/5 (60.0%) | 3/4 (75.0%) | 1/5 (20.0%) | 0.42 |  |
| Previous tuberculosis treatment |  |  |  |  |  |
| - Yes | 1/2 (50.0%) | 1/1 (100%) | 0/1 (0%) | 1.00 | 0.28 |
| - No | 10/36 (27.8%) | 7/35 (20.0%) | 6/37 (16.2%) | 0.51 |  |
| Drug susceptibility* |  |  |  |  |  |
| - MDR-TB | 0/0 (0%) | 1/1 (100%) | 0/0 (0%) | 1.00 | 0.17 |
| - Rifampicin mono-resistance | 0/1 (0%) | 0/0 (0%) | 0/0 (0%) |  |  |
| - Isoniazid resistance (with or without streptomycin resistance) | 3/6 (50.0%) | 1/1 (100%) | 1/2 (50.0%) |  |  |
| - No or other resistance | 5/10 (50.0%) | 1/7 (14%) | 1/12 (8%) | 0.09 |  |
| LTA4H genotype† |  |  |  |  |  |
| - CC | 3/18 (16.7%) | 4/13 (30.8%) | 0/16 (0%) | 0.05 | 0.13 |
| - CT | 7/16 (43.8%) | 3/16 (18.8%) | 5/19 (26.3%) | 0.30 |  |
| - TT | 1/4 (25.0%) | 1/6 (16.7%) | 0/1 (0%) | 1 |  |

*4 participants not genotyped. P-value for the heterogeneity was obtained from likelihood ratio tests for an interaction term between treatment and the grouping variable in a logistic regression model.

†Outcomes unavailable in two participants in the placebo group with DST available (one lost to follow-up at day 57 and one did not have MRI information at day 60).

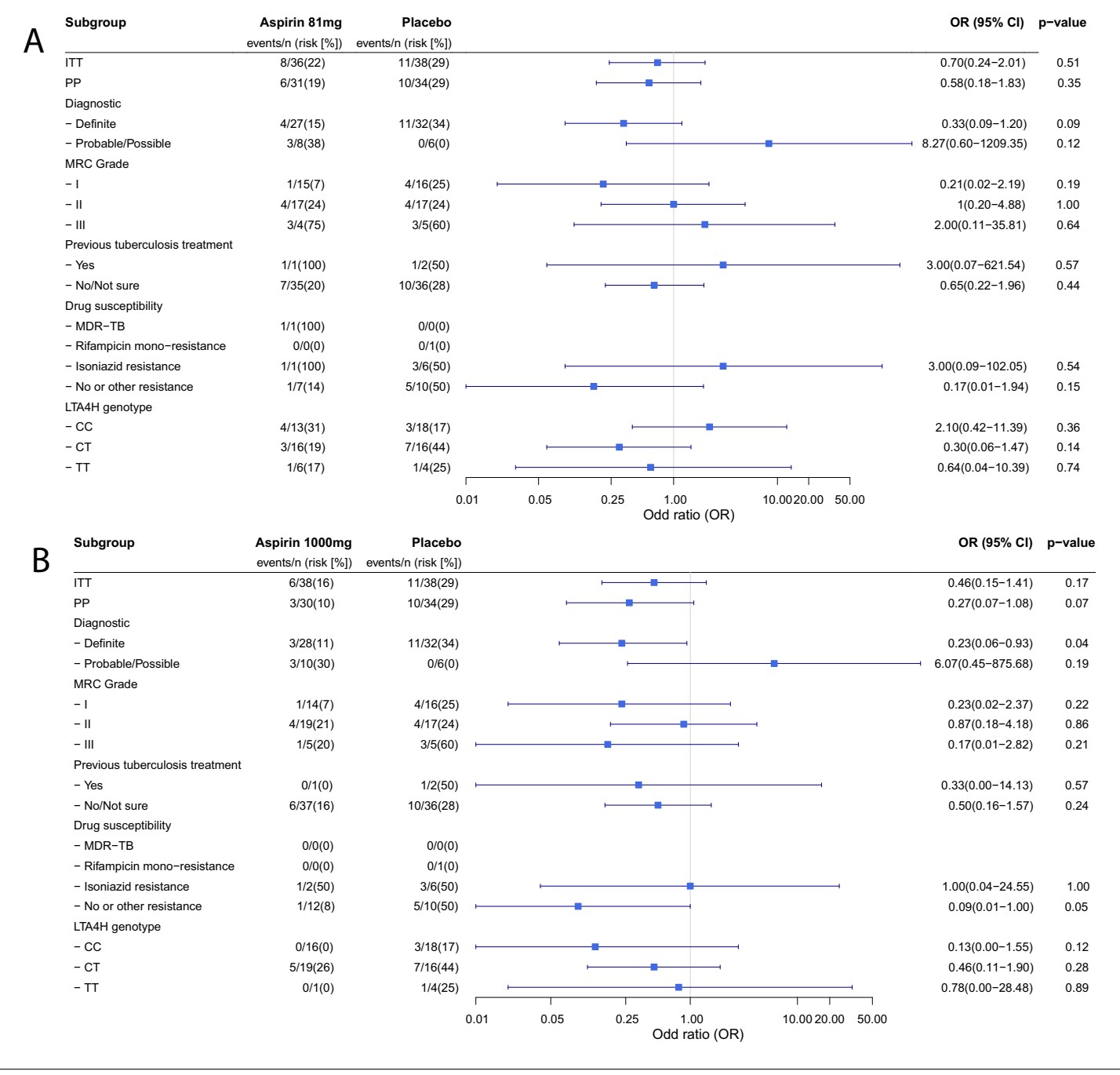

**Figure 2.** Forest plots of ITT, per-protocol and planned sub-group analysis of aspirin 81 mg versus placebo (A) and aspirin 1000 mg versus placebo (B) for the primary efficacy outcome. Estimates for subgroups without events were obtained via Firth's penalized likelihood. Panels show 8 month survival plots for the ITT (C) and per-protocol (D) populations.

The online version of this article includes the following figure supplement(s) for figure 2:

**Figure supplement 1.** C (panel within *Figure 2*).
**Figure supplement 2.** D (panel within *Figure 2*).

those treated with placebo (33 (80.5%)) than aspirin 81 mg (24 (61.5%)) or 1000 mg (25 (62.5%)) (p=0.11).

**Table 5.** Summary of disability and death by day 60 and by 8 months in the ITT and per-protocol populations

| | Placebo (N = 41) | Aspirin 81 mg (N = 39) | Aspirin 1000 mg (N = 40) | P-value |
|---|---|---|---|---|
| **ITT population** | | | | |
| **Rankin score categories by 60 days*** | | | | |
| - Complete recovery | 6/37 (16.2%) | 11/39 (28.2%) | 9/40 (22.5%) | |
| - Intermediate | 22/37 (59.5%) | 16/39 (41.0%) | 22/40 (55.0%) | 0.61 |
| - Death or severely disabled | 9/37 (24.3%) | 12/39 (30.8%) | 9/40 (22.5%) | |
| **Rankin score categories by 8 months** | | | | |
| - Complete recovery | 18/39 (46.1%) | 23/39 (59.0%) | 22/40 (55.0%) | |
| - Intermediate | 10/39 (24.6%) | 7/39 (17.95%) | 11/40 (27.5%) | 0.29 |
| - Death or severely disabled | 11/39 (28.2%) | 9/39 (23.1%) | 7/40 (17.5%) | |
| **Per-protocol population** | | | | |
| **Rankin score categories at 60 days** | | | | |
| - Complete recovery | 6/33 (18.2%) | 9/31 (29.0%) | 7/31 (22.6%) | |
| - Intermediate | 21/33 (63.6%) | 15/31 (48.4%) | 19/31 (61.3%) | 0.69 |
| - Death or severely disabled | 6/33 (18.2%) | 7/31 (22.6%) | 5/31 (16.1%) | |
| **Rankin score categories by 8 months** | | | | |
| - Complete recovery | 17/35 (48.6%) | 20/31 (64.5%) | 19/31 (61.3%) | |
| - Intermediate | 9/35 (25.7%) | 5/31 (16.1%) | 9/31 (29.0%) | 0.13 |
| - Death or severely disabled | 9/35 (25.7%) | 6/31 (19.4%) | 3/31 (9.7%) | |

*Complete recovery = Rankin score 0; Intermediate = Rankin score 1 or 2; Death or severely disabled = Rankin score 3–6. Three participants in the placebo group missed their day 60 assessment and two were lost to follow-up at day 57 and 217. P-values refer to a linear-by-linear trend test for disability scores

## CSF lipid mediator profiles

We investigated the impact of aspirin on the concentrations of lipid mediators of inflammation, extracted, identified and quantified from CSF using lipid mediator profiling and LC-MS/MS (*Figure 3 - child*). Partial least squared discriminant analysis 2-dimensional score plot of CSF taken from all surviving participants 30 days from randomization showed clustering of lipid mediators according to treatment group suggesting dose-dependent effects (*Figure 3A*). Furthermore, in those participants who received >30 days of study drug we compared baseline with day 30 CSF and found dose-dependent inhibition of $TXB_2$ (the stable metabolite of $TXA_2$) and up-regulation of pro-resolving protectins, with significant differences observed in the aspirin 1000 mg group compared to placebo (*Figure 3B*; *supplementary file 6*).

## Discussion

There is much current interest in novel host-directed therapies against tuberculosis (*Wallis et al., 2016*). We conducted a phase two randomised controlled trial with the aim of showing the safety and potential efficacy of either low (81 mg/day) or higher (1000 mg/day) dose aspirin when added to anti-tuberculosis drugs and dexamethasone for the treatment of HIV-uninfected adults with TBM. We found that aspirin was not associated with a significant increase in grade 3 or four adverse events. In both the ITT and the per-protocol population, the observed risk of death or new brain infarction by day 60 was lower in the aspirin arms compared to placebo, although this was not statistically significant. Planned sub-group analyses, however, suggested that aspirin 1000 mg may benefit those with microbiologically confirmed TBM. This finding was supported by an analysis of CSF lipid mediators of inflammation, which demonstrated aspirin 1000 mg was associated with significant inhibition of pro-thrombotic $TXA_2$ and upregulation of pro-resolution protectins.

The important characteristics of the trial participants, which influences the generalisability of the findings, were that they were HIV-uninfected, had relatively mild disease (87.5% MRC grade I or II), a high proportion (35.1%) had brain infarcts evident at baseline, and most (76.7%) had a microbiologically confirmed diagnosis of TBM. The high proportion of microbiologically confirmed disease is

**Table 6.** Summary of clinical grade 3 or four adverse events by randomised group

| Event | Placebo (n = 41) No. patients (%) (number of events) | Aspirin 81 mg (n = 39) No. patients (%) (number of events) | Aspirin 1000 mg (n = 40) No. patients (%) (number of events) | P-value comparison |
|---|---|---|---|---|
| All events | 11 (26.8%) (17) | 17 (43.6%) (33) | 9 (22.5%) (16) | 0.11 |
| Allergic events<br>- Rash<br>- Stevens Johnsons syndrome* | 1 (2.4%) (1)<br>1 (2.4%) (1)<br>0 | 1 (2.6%) (2)<br>0<br>1 (2.6%) (2) | 0 | 0.77 |
| Cardiac events<br>- Hypotension | 0 | 3 (7.7%) (3)<br>3 (7.7%) (3) | 1 (2.5%) (1) | 0.08 |
| Electrolyte events<br>- Hyponatraemia<br>- Hypokalaemia | 1 (2.4%) (1)<br>1 (2.4) (1)<br>0 | 4 (10.3%) (4)<br>4 (10.3%) (4)<br>0 | 3 (7.5%) (3)<br>2 (5.0%) (2)<br>1 (2.5%) (1) | 0.31 |
| Gastrointestinal events<br>- Vomiting blood<br>- Melena | 2 (4.9%) (2)<br>2 (4.9%) (2)<br>0 | 1 (2.6%) (1)<br>1 (2.6%) (1)<br>0 | 1 (2.5%) (1)<br>0<br>1 (2.5%) (1) | 1.00 |
| Hepatic events<br>- Hepatitis | 1 (2.4%) (2)<br>1 (2.4%) (2) | 1 (2.6%) (1)<br>1 (2.6%) (1) | 0 | 0.77 |
| Neurological events<br>- Hemiparesis<br>- Paraparesis<br>- Cranial nerve palsy<br>- Fall in GCS ≥ 2 points for ≥ 2 days | 4 (9.8%) (5)<br>0<br>1 (2.4%) (1)<br>0<br>2 (4.9%) (2) | 9 (23.1%) (13)<br>1 (2.6%) (1)<br>2 (5.1%) (2)<br>1 (2.6%) (1)<br>5 (12.8%) (5) | 3 (7.5%) (5)<br>0<br>1 (2.5%) (1)<br>0<br>2 (5.0%) (2) | 0.11 |
| Respiratory events<br>- Pneumonia<br>- Respiratory failure | 4 (9.8%) (4)<br>0<br>4 (9.8%) (4) | 8 (20.5%) (9)<br>1 (2.6%) (1)<br>8 (20.5%) (1) | 4 (10.0%) (4)<br>0<br>4 (10.0%) (4) | 0.30 |
| Other events† | 2 (4.9%) (2) | 0 | 2 (5.0%) (2) | 0.54 |

*Event related to rifampicin, not study drug

P-values refer to Fisher's exact test for the number of participants with at least one event.

especially relevant, as many centres report much lower proportions (typically 20–50%). The characteristics of populations of suspected but unconfirmed case of TBM may vary substantially between centres and influence treatment effects. In addition, all participants were treated with adjunctive dexamethasone, which is known to reduce deaths from TBM in this population (*Thwaites et al., 2004*).

Combining dexamethasone with aspirin did not significantly increase gastro-intestinal bleeding of any severity, or any other category of grade 3 or four adverse event. There was a non-significant increase in non-severe (grade 1 or 2) gastro-intestinal bleeding events (mostly small volumes of digested blood aspirated from nasogastric tubes) in the aspirin-treated participants and in all these cases the study drug was stopped immediately. A larger trial is needed to determine whether aspirin truly increases these events and to assess their clinical significance. However, as the risk of severe gastric bleeding appears to be very low, the future management of minor bleeding events in aspirin-treated patients could be less conservative, especially as the per-protocol analysis suggested participants who received >30 days of aspirin may benefit more than those who stopped aspirin earlier.

Our findings are similar to the previous trial of aspirin (150 mg/day) for adults with TBM conducted in India (*Misra et al., 2010*), which reported aspirin in combination with prednisolone was associated with a reduction in brain infarcts (22% versus 55% in controls; p=0.08) and death (13% verus 14% in controls; p=0.05). Treatment with aspirin without prednisolone was not associated with improved outcomes. In both the Indian trial and the trial conducted in South African children (*Schoeman et al., 2011*), which included a high dose (100 mg/kg/day) arm, the use of aspirin was not associated with increased adverse events. In particular, aspirin did not appear to increase the risk of gastro-intestinal bleeding in either study. If these data are taken together with the results of the current study, they strongly suggest aspirin at doses ranging from 81 mg to 1000 mg per day can be safely added to anti-tuberculosis and corticosteroid therapy. Determining which dose is likely

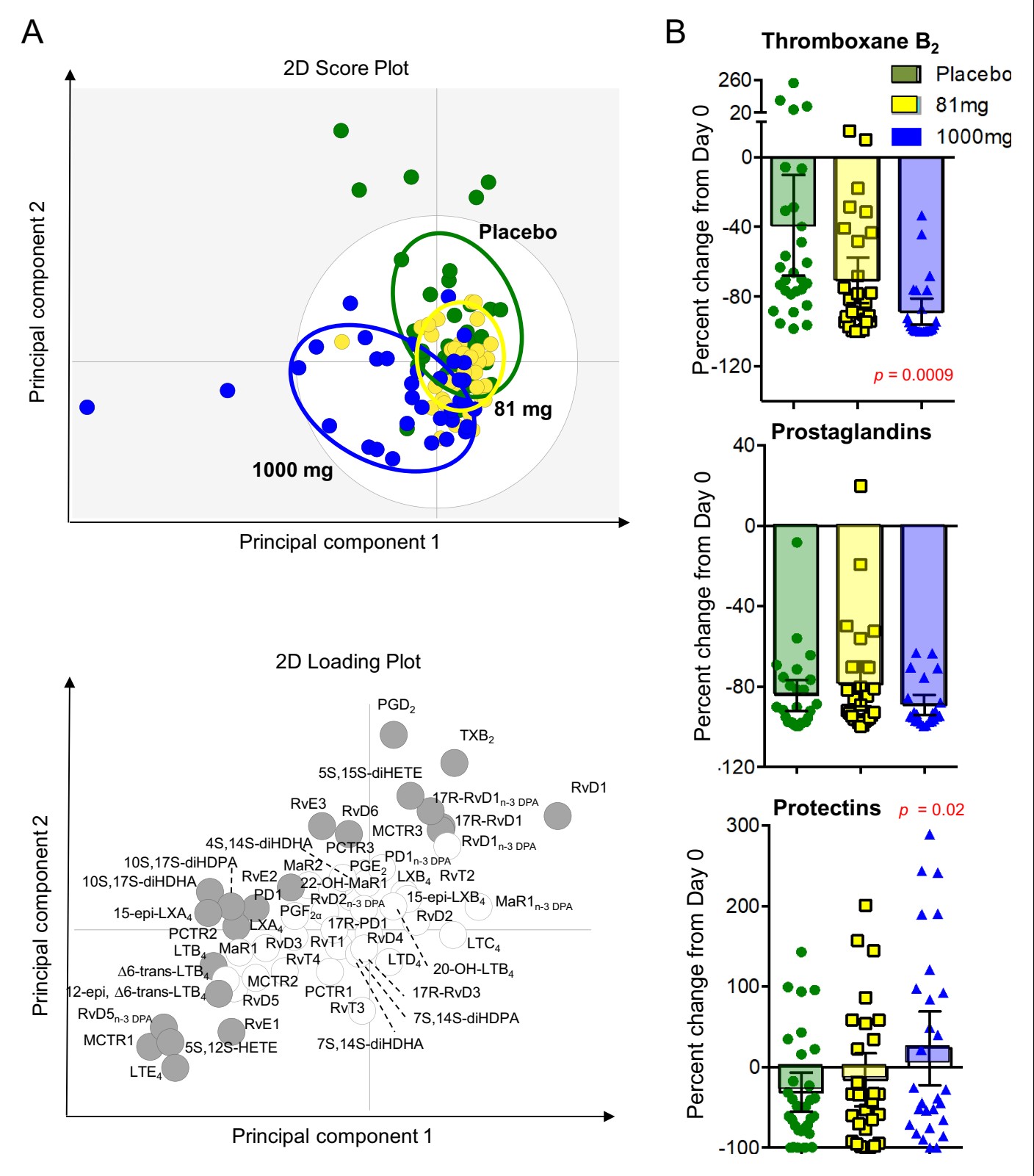

**Figure 3.** LCMS lipid mediator profiles in the CSF of adults with TBM according to treatment with aspirin or placebo. CSF was collected from participants at baseline and 30 days after 81 mg, 1000 mg or placebo administration. (A) Partial least squares discriminant analysis 2-dimensional score plot of the distinct LM-SPM profiles identified in day 30 CSF at the indicated intervals (*top panel*) and corresponding 2-dimensional loading plot. Grey ellipse in the score plots denotes estimated 95% probability regions (*bottom panel*). Grey circles in the loading plot represent LM with a variable in

*Figure 3 continued on next page*

*Figure 3 continued*

importance score $\geq 1$. (**B**) Relative regulation of Thromboxane $B_2$ (the stable $TXA_2$ further metabolite), Prostaglandins ($PGD_2$, $PGE_2$, $PGF_2$) and Protectins (PD1, 17R-PD1, 22-OH-PD1, 10S, 17S-diHDHA, PCTR1, PCTR2 and PCTR3) by day 30 compared to baseline values (absolute values given in *supplementary file 6*). Results for **B** are mean ± s.e.m, n = 30 for placebo, n = 26 for 81 mg and n = 26 for 1000 mg group. Comparisons between treatment groups assessed using one-way ANOVA followed by multiple comparisons test. Only p-values<0.05 given in the figure (all other comparisons non-significant).

The online version of this article includes the following figure supplement(s) for figure 3:

**Figure supplement 1.** Lipid mediator profiles of CSF from participants with TBM.

---

to be most effective is difficult, but our findings suggest that higher doses (1000 mg/day or equivalent in children) are likely to more effective.

The strength of our trial is that it addressed both the potential clinical role of aspirin and its mechanism of action by serial brain imaging and analysis of a panel of 71 lipid mediators, their precursors, pathway markers and further metabolites in the CSF. There are, however, some important limitations. First, assessment of the primary efficacy endpoint required participants to be well enough to have an MRI at baseline and day 60. Three screened patients were judged too unwell to have baseline imaging and enter the trial, and five participants were too unwell for imaging at day 60. Therefore, our findings may not be generalisable to those with very severe disease at baseline.

The trial was not powered to show an impact on longer term survival or neurodisability and therefore does not provide definitive, practice-changing evidence that adjunctive aspirin improves outcomes in all adults with TBM. However, the findings support the hypothesis that aspirin has effects on tuberculosis-associated neuro-inflammation that are independent of dexamethasone and may lead to additional improvements in clinical outcomes. The clinical findings need to be interpreted cautiously, but the planned sub-group analysis suggested a clinical benefit of aspirin in those with microbiologically confirmed disease, especially at 1000 mg. This potential clinical effect is supported by the CSF analysis, which showed dose-dependent inhibition of $TXA_2$ by aspirin, modest inhibition of prostaglandins, and the upregulation of potentially protective protectins. The serial brain images also support the assertion that aspirin's benefit may be driven by the upregulation of these pro-resolving molecules: 60% of infarcts seen at baseline had resolved by day 60 in the aspirin 1000 mg group compared with 14.2% in the aspirin 81 mg group and 42.9% in the placebo group.

In summary, this phase two randomised placebo-controlled trial suggests that daily aspirin 81 mg or 1000 mg can be given safely with dexamethasone and anti-tuberculosis drugs for the treatment of HIV-uninfected adults with TBM. The trial also provides new data that indicate aspirin induces dose-dependent inhibition of $TXA_2$ and upregulation of protectins within the central nervous system that may reduce the incidence and promote the resolution of TBM-associated brain infarcts and inflammation and thereby improve outcome. These findings provide strong support for the conduct of a large phase 3 trial of adjunctive aspirin for TBM, but may also have relevance for the treatment of other forms of tuberculosis, adding to the growing evidence that aspirin and other non-steroidal anti-inflammatory drugs may be useful novel adjunctive agents in tuberculosis treatment (*Kroesen et al., 2017*).

## Materials and methods

### Study design

We conducted a parallel group, double blind, randomised, placebo controlled trial in HIV-uninfected adults with TBM to assess the safety and efficacy of either 81 mg or 1000 mg aspirin daily for the first 60 days of treatment with standard anti-tuberculosis drugs and dexamethasone (full study protocol provided in *supplementary file 7*). The trial enrolled in-patients at the Hospital for Tropical Diseases, a 550-bed tertiary referral hospital in Ho Chi Minh City, Vietnam. The trial was approved by the Oxford Tropical Research Ethics Committee and the Institutional Review Board of the Hospital for Tropical Diseases and the Ethical Committee of the Ministry of Health, Vietnam.

## Participants

Adults (≥18 years old) with suspected TBM (at least 5 days of meningitis symptoms, nuchal rigidity, and CSF abnormalities) and a negative HIV test were eligible to enter the trial. Written informed consent to participate in the study was obtained from all participants or from their relatives if the participant could not provide consent due to incapacity. Published diagnostic criteria were used to categorise participants retrospectively into definite, probable, or possible TBM once the results of all investigations returned (*supplementary file 1*) (*Marais et al., 2010*).

Patients were excluded if they or their family did not give written informed consent to participate; they had received >2 days anti-tuberculosis chemotherapy for the current infection; they were unlikely, for any reason, to be able to have MRI brain imaging within 5 days of randomisation; they had known or suspected infection with multi-drug resistant (MDR) tuberculosis (resistant to at least isoniazid and rifampicin); they were unable to take isoniazid, rifampicin, or pyrazinamide at recommended doses for any reason; they had a history of peptic ulceration or gastro-intestinal bleeding, or active gastro-intestinal bleeding was suspected; they had taken >1 dose of aspirin (at any dose) or any other NSAID for any reason within 2 weeks of screening; aspirin was considered mandatory for any reason; dexamethasone was contraindicated for any reason; or the patient was pregnant or breast feeding.

## Randomisation and blinding

Randomisation was 1:1:1 to placebo, 81 mg aspirin, or 1000 mg aspirin according to a computer-generated randomization list using block randomization with variable blocks of length 3 (with 25% probability) and 6 (with 75% probability) and with stratification by MRC disease severity grade (defined in *Table 1*). Participant numbers were stratified and assigned sequentially at randomisation, with each participant receiving a pre-prepared numbered identical package of blinded study drugs. Treatment allocation was concealed by each treatment pack containing two bottles of study treatment: one containing 81 mg tablets of identical aspirin or placebo, the second containing identical 500 mg tablets of aspirin or placebo. Trial participants and the entire clinical and study team were blind to the treatment allocation for the duration of the trial. For 60 days, participants took one 81 mg tablet/placebo and two 500 mg tablets/placebo (taken every 12 hr), according to their randomised allocated treatment. Participants unable to swallow were given crushed tablets (which did not reveal the allocation) via a nasogastric tube at the same doses.

## Procedures

Anti-tuberculosis treatment followed Vietnam's tuberculosis treatment guidelines, consisting of oral isoniazid (5 mg/kg/day; maximum 300 mg/day), rifampicin (10 mg/kg/day), pyrazinamide (25 mg/kg/day; maximum 2 g/day) and ethambutol (20 mg/kg/day; maximum 1.2 g/day) and intramuscular streptomycin (20 mg/kg/day; maximum 1 g/day) for 3 months, followed by rifampicin and isoniazid at the same doses for a further 6 months. All patients received adjunctive dexamethasone for the first 6 to 8 weeks of treatment as previously described (*Thwaites et al., 2004*) and oral ranitidine (300 mg at night). For patients infected with *M. tuberculosis* resistant to isoniazid, the treatment was adjusted according to local guidelines and the susceptibility of the organism.

Lumbar puncture was performed before the start of treatment and on days 30 and 60 as per normal clinical care. All CSF specimens were stained and cultured by standard methods for pyogenic bacteria, fungi, and mycobacteria and tested by GeneXpert MTB/RIF assay (Cepheid, USA). Isolates of *Mycobacterium tuberculosis* were tested for susceptibility to isoniazid, rifampicin, ethambutol and streptomycin by mycobacterial growth indicator tube method (*Ardito et al., 2001*).

Brain MRI (1.5 tesla; 64 slices) with T1 volume pre and post contrast, T2, FLAIR, gradient echo and DWI sequences was acquired ±5 days from randomisation and then at day 60 (±10 days) and day 240 (±30 days). All images were separately interpreted by two independent neuroradiologists (one consultant and one fellow) blind to the treatment allocation, who then determined a consensus opinion.

Clinical progress and neurological and drug-related adverse events were assessed daily until discharge from hospital. After discharge, monthly visits were scheduled for clinical evaluation and laboratory monitoring until 8 months, when a final clinical assessment was made. All participants were genotyped for leukotriene A4 hydrolase (LTA4H), which has been shown to influence TBM

pathophysiology and outcome, using previously described methods to investigate its influence on CSF inflammation and aspirin effect (*Thuong et al., 2017*).

## Outcomes

The primary safety endpoint was the occurrence of clinically significant upper gastro-intestinal bleeding and any cerebral bleeding confirmed by brain imaging by 60 days from randomisation. Clinically significant upper gastro-intestinal bleeding was defined as vomiting fresh or changed blood of any volume; passing melena; an unexplained drop in haemoglobin concentration of >2 g/L; or >5 mls of fresh or changed blood aspirated from nasogastric tube. The primary efficacy endpoint was any new MRI-proven brain infarction or death by 60 days.

The secondary endpoints were the number of grade 3 and four and serious adverse events by day 60 from randomisation; mortality over the first 240 days from randomisation; duration of hospital stay; neurological disability (as assessed by the modified Rankin score) by days 60 and 240; the proportion of patients with MRI-proven infarction by day 240; and the resolution of CSF inflammation by day 30 through measurement of lipid mediators.

## CSF lipid mediator profiling

Lipid mediators were measured by liquid Chromatography-tandem mass spectrometry (LC-MS/MS) on baseline and day 30 CSF (archived at −80℃). Methods have been previously described, (*Walker et al., 2017*) and are briefly summarised here.

Baseline and day 30 CSF (archived at −80℃) was placed in two volumes ice-cold methanol containing deuterium labelled $PGE_2$ ($d_4$-$PGE_2$); $d_4$-$LTB_4$, $d_5$-$LXA_4$ $d_5$-RvD2, $d_5$-$LTC_4$, $d_5$-$LTD_4$, and $d_5$-$LTE_4$ (500 pg each; Cayman Chemicals). These were kept at −20℃ for 45 min to allow for protein precipitation and lipid mediators were extracted using C-18 based Solid Phase Extraction as previously described (*Walker et al., 2017*). Methyl formate fractions were brought to dryness using a TurboVap LP (Biotage) and products suspended in water-methanol (80:20 vol:vol) for Liquid Chromatography-tandem mass spectrometry (LC-MS/MS) based profiling. A Shimadzu LC-20AD HPLC and a Shimadzu SIL- 20AC autoinjector (Shimadzu, Kyoto, Japan), paired with a QTrap 5500 (ABSciex, Warrington, UK) were utilised and operated as previously described (*Colas et al., 2014*). To monitor each lipid mediator and deuterium labelled internal standard, a Multiple Reaction Monitoring method was developed using parent ions and characteristic diagnostic ion fragments (*Walker et al., 2017*). This was coupled to an Information Dependent Acquisition and an Enhanced Product Ion scan. Identification criteria included matching retention time to synthetic standards and at least six diagnostic ions in the MS-MS spectrum for each molecule. Calibration curves were obtained for each molecule using authentic compound mixtures and deuterium labelled lipid mediator at 0.78, 1.56, 3.12, 6.25, 12.5, 25, 50, 100, and 200 pg. Linear calibration curves were obtained for each lipid mediator, which gave r (*Ruslami et al., 2013*) values of 0.98–0.99.

## Statistical analysis

The sample size of 40 per arm was chosen based on clinical and feasibility considerations. Our objective was to provide estimates of safety and efficacy, alongside potential mechanisms of action, which would inform the design and execution of a larger phase III trial; we did not expect to show clinically definitive efficacy in this phase two trial. We assumed a risk of new MRI-proven brain infarction or death within 60 days of approximately 40% in the control arm. Based on the results of the trial performed by Misra et al, (*Misra et al., 2010*) we assumed that 81 mg aspirin daily may reduce this risk to 20% and the risk in the 1000 mg aspirin daily arm to between 20–40%. Given these estimates our three arm trial would have approximately 75% power to detect such an effect at the one-sided 10% significance level (i.e. to generate 'mild evidence') and approximately 50% power to detect it at the conventional one-sided 2.5% significance level.

Cerebral bleeding associated with TBM is extremely rare (estimated at <0.01% of all patients), but it is possible it may be more common in those treated with aspirin. It was therefore included in the primary assessment of safety. The proportion of HIV-uninfected patients with clinically significant gastro-intestinal bleeding from our most recent trial of hyper-intensive anti-tuberculosis chemotherapy in Vietnam was about 1% (3/278) (*Heemskerk et al., 2016*). Assuming this same risk of bleeding in the aspirin arms, the probability that the *upper* limit of the 95% confidence interval remains below

12.9% is more than 92%. If the risk of bleeding is 10% in one of the aspirin arms, the probability that the *lower* limit of the 95% confidence interval remains above 2.5% is 78%.

Statistical analysis followed the protocol and a predefined statistical analysis plan. The main population for all analyses was the intention-to treat population (ITT), which included all randomized participants, analysed according to the randomized treatment arm. A per-protocol population was defined by excluding participants with a confirmed alternative diagnosis to TBM; those with confirmed MDR-TBM; and those who received <30 days of study drug for reasons other than death.

The risk of clinically significant upper-gastro-intestinal bleeding and image-proven cerebral bleeding by 60 days were summarized as numbers and proportion in each group together with two-sided 95% confidence intervals for risk differences between groups calculated by the Wilson score method. Comparison between the three arms was based on the chi-square test of independence. The primary efficacy endpoint of new MRI-proven brain infarction or death by 60 days was analysed in the same way. For the secondary endpoint, a linear-by–linear trend test (also called Cochran–Armitage test for trend) was performed to assess the association between disability score and the three arms (*Agresti, 2002*).

Subgroup analyses for the primary efficacy endpoint were conducted using logistic regression in pre-defined subgroups according to TBM grade (I, II, or III); previous tuberculosis treatment; TBM diagnostic category (definite versus probable/possible); drug resistance (MDR-TB, rifampicin mono-resistance, isoniazid resistance (with or without streptomycin resistance), no or other resistance); and leukotriene A4 hydrolase (LTA4H) genotype (CC, CT, TT). We fitted a logistic regression model using Firth's correction to the likelihood because there were subgroups without events (*Firth, 1993*). Heterogeneity of the treatment effect across sub-groups was tested via a likelihood ratio test for the interaction term between treatment and the grouping variable in a logistic regression model.

To assess differences of lipid mediators between the different treatment groups partial least squares discriminant analysis was conducted using SIMCA 13.0.3 software (Umetrics, San Jose, CA), (*Colas et al., 2014*) where mediators displaying a Variable Importance in Projection scores greater than one were taken as displaying significant correlation with the treatment group. This parameter estimates the importance of a variable in the Partial Least Squares projections with scores greater than one indicating that a specific variable is important in a given model.

No imputation of missing data was performed and the threshold for assuming statistical significance was $p<0.05$ for all analyses. For the analysis of gastro-intestinal/cerebral bleeding by day 60, patients lost to follow-up or dead before day 60 (without prior bleeding) were excluded from the analysis to avoid under estimating the true proportion of bleeding in the aspirin groups. The analysis of the primary efficacy endpoint excluded patients with missing baseline or follow-up MRI scans (except for prior death). All statistical analyses were performed with the statistical software R v3.1.2 (*R Core Team, 2017*).

An independent data and safety monitoring board reviewed the unblinded safety data after 39 and 94 participants were enrolled. The trial was not stopped early. The trial was registered on clinicaltrials.gov, NCT02237365.

## Role of the funding source

The funders played no part in the design, implementation, or analysis of the study or in the decision to publish the results. The corresponding author has full access to all the data in the study and had final responsibility for the decision to submit for publication.

## Data availability

The Oxford University Clinical Research Unit (OUCRU) operates managed open access to the research data it generates, which complies with the policies of its major funder, the Wellcome Trust, UK. The objective is not to restrict access to data, but to monitor who uses the data and for what purpose, and to ensure those responsible for collecting and curating the data are appropriately acknowledged by those using it. Therefore, those wishing to acquire the anonymized dataset, including LC-MS/MS data, from which the results presented in this manuscript were produced should email the trial Chief Investigator and corresponding author, Professor Guy Thwaites (gthwaites@oucru.org).

## Acknowledgements

We thank the independent data and safety monitoring board, Professor Sarah Walker (Chair), Professor Graeme Meintjes, Dr Nguyen Viet Nhung, Dr Le Van Nhi, and Dr Le Minh. The study was funded by the Wellcome Trust, UK (106680/Z/14/Z to GT); and the European Research Council (grant no: 677542) and St. Bartholomew's Charity (grant no: MGU0343) and Wellcome Trust/Royal Society Henry Dale Fellowship (107613/Z/15/Z) to JD.

## Additional information

### Funding

| Funder | Grant reference number | Author |
|---|---|---|
| H2020 European Research Council | 677542 | Jesmond Dalli |
| St Bartholomews Charity | MGU0343 | Jesmond Dalli |
| Wellcome Trust | 110179/Z/15/Z | Guy E Thwaites |
| Wellcome Trust | 106680/Z/14/Z | Guy E Thwaites |

The funders had no role in study design, data collection and interpretation, or the decision to submit the work for publication.

### Author contributions

Nguyen TH Mai, Conceptualization, Data curation, Investigation, Project administration, Writing—review and editing; Nicholas Dobbs, Data curation, Formal analysis, Investigation, Visualization, Writing—review and editing; Nguyen Hoan Phu, Conceptualization, Data curation, Supervision, Investigation, Methodology, Project administration, Writing—review and editing; Romain A Colas, Resources, Formal analysis, Investigation, Methodology, Writing—review and editing; Le TP Thao, Data curation, Formal analysis, Validation, Methodology, Writing—original draft, Writing—review and editing; Nguyen TT Thuong, Conceptualization, Formal analysis, Investigation, Project administration, Writing—review and editing; Ho DT Nghia, Evelyne Kestelyn, Data curation, Supervision, Investigation, Project administration, Writing—review and editing; Nguyen HH Hanh, Nguyen T Hang, A Dorothee Heemskerk, Data curation, Investigation, Project administration, Writing—review and editing; Jeremy N Day, Conceptualization, Investigation, Methodology, Writing—review and editing; Lucy Ly, Resources, Data curation, Formal analysis, Investigation, Methodology, Writing—review and editing; Do DA Thu, Resources, Data curation, Investigation, Visualization, Writing—review and editing; Laura Merson, Data curation, Supervision, Investigation, Methodology, Project administration, Writing—review and editing; Marcel Wolbers, Formal analysis, Supervision, Validation, Investigation, Methodology, Writing—original draft, Writing—review and editing; Ronald Geskus, Data curation, Formal analysis, Supervision, Validation, Investigation, Methodology, Writing—original draft, Writing—review and editing; David Summers, Formal analysis, Supervision, Investigation, Visualization, Methodology, Writing—review and editing; Nguyen VV Chau, Conceptualization, Supervision, Investigation, Methodology, Project administration, Writing—review and editing; Jesmond Dalli, Conceptualization, Resources, Data curation, Formal analysis, Supervision, Funding acquisition, Validation, Investigation, Methodology, Writing—original draft, Project administration, Writing—review and editing; Guy E Thwaites, Conceptualization, Supervision, Funding acquisition, Investigation, Methodology, Writing—original draft, Project administration, Writing—review and editing

### Author ORCIDs

Laura Merson  https://orcid.org/0000-0002-4168-1960
Guy E Thwaites  http://orcid.org/0000-0002-2858-2087

### Ethics

Clinical trial registration Registered on clinicaltrials.gov, NCT02237365.

Human subjects: Written informed consent to participate in the study was obtained from all participants or from their relatives if the participant could not provide consent due to incapacity. The trial was approved by the Oxford Tropical Research Ethics Committee and the Institutional Review Board of the Hospital for Tropical Diseases and the Ethical Committee of the Ministry of Health, Vietnam.

### Decision letter and Author response
Decision letter https://doi.org/10.7554/eLife.33478.sa1
Author response https://doi.org/10.7554/eLife.33478.sa2

## Additional files

### Supplementary files
• Supplementary file 1. Table S1 Tuberculous meningitis diagnostic criteria

• Supplementary file 2. Table S2 Reasons for exclusion of screened patients from the trial

• Supplementary file 3. Table S3 Summary of adverse events related, or possibly related, to aspirin

• Supplementary file 4. Table S4 Full Rankin scores by treatment group by day 60 and 8 months in the ITT population

• Supplementary file 5. Table S5 Other MRI brain findings by treatment group on days 60 and month eight in the ITT population

• Supplementary file 6. Table S6 CSF individual lipid mediator Profiles at baseline and 30 days post aspirin and placebo administration†

• Supplementary file 7. Full trial protocol.

• Reporting standard

• Transparent reporting form

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
