## [Decision Letter]

Thank you for submitting your article "A randomised double blind placebo controlled phase 2 trial of adjunctive aspirin for tuberculous meningitis" for consideration by *eLife*. Your article has been reviewed by five peer reviewers, and the evaluation has been overseen by Madhukar Pai, a Reviewing Editor, and Michel Nussenzweig as the Senior Editor. The following individuals involved in review of your submission have agreed to reveal their identity: Michael Marks (Reviewer #1); Shriprakash Kalantri (Reviewer #3); Patrick Phillips (Reviewer #5).

The reviewers have discussed the reviews with one another and the Reviewing Editor has drafted this decision to help you prepare a revised submission.

Summary:

The authors present data from a phase 2 RCT on use of aspirin to improve outcomes in Tuberculosis Meningitis. Given the poor outcomes in this group additional interventions are urgently needed for TBM. This is a well conducted RCT presenting important information which may inform future phase 3 trials. The modest conclusion of the authors is warranted: that aspirin is likely to be safe, but a larger phase 3 trial should be conducted to fully assess efficacy. In summary, this paper provides helpful data is an important area and paves the way for (and arguably mandates) a larger efficacy trial.

Essential revisions:

1) No CONSORT checklist has been provided, and, for instance, the Abstract does not follow the structure recommended by CONSORT. Kindly make sure the revision follows the CONSORT format and also provide the checklist.

2) One method issue I have is that the method of allocation concealment was not mentioned in the Materials and methods section. The second is that the sample size calculation was over ambitious and has led to a small sample size and the results of this trial are now affected significantly by β error. This affects both the safety outcome of bleeding (which was more in both aspirin arms) as well as efficacy that was better in the aspirin arms. This needs to be reflected in their Discussion. With the small sample and multiple subgroup comparisons the authors need to exercise caution in the interpretation of their results. Given these concerns I feel the final conclusions should reflect *no* difference in primary safety or efficacy outcomes with possibility of increased harms (bleeding) as well as benefit, which needs a larger trial.

3) The section on sample size calculation is a bit difficult to understand. I suggest that the authors reframe this section. Here is a template that they may use to help the audience understand the trial.

We calculated that with a sample size of at least ---- patients, the trial would have 80% power to detect a ----percentage-point lower risk of death among patients receiving aspirin treatment than among those receiving the standard treatment (---% vs. ---%, corresponding to a target hazard ratio of ---) at a two-sided 5% significance level.

4) The main limitation is that the trial is small and underpowered even for the primary outcomes. This is a shame, as the results are promising and a slightly larger study might have shown more convincing results. The section describing the sample size is hard to follow, and would benefit from text describing more clearly what differences the study was designed to show, and to expand more on the 'clinical and feasibility considerations' that limited the sample size. Given the small sample size, and the many hypothesis tests conducted, and the resulting multiplicity considerations, the authors need to be careful not to over-state the results. Language needs to be more consistent as to what is considered evidence of a difference and what is just an observed statistically non-significant difference. There are places in the text where a difference is declared even though the p-value is greater than the common (nominal) cut-off of 0.05, e.g. subsection “Primary outcomes”, third paragraph (p=0.18) and subsection “Secondary outcomes and adverse events”, last paragraph (p=0.11). The sample size calculations talk about 'mild evidence' with a 10% one-sided significance level. If this is the threshold for evidence that the authors are using, this should be clearly stated in the Materials and methods.

Although pre-planned, there were 5 different sub-group analyses, and the authors found one sub-group that showed a significant difference, albeit p=0.06. The authors should comment on the issues regarding multiplicity that arise and how this impacts on the interpretation of this results. There is a clear biological rationale for this particular sub-group so the results are compelling nevertheless.

---

## [Author Response]

Essential revisions:1) No CONSORT checklist has been provided, and, for instance, the Abstract does not follow the structure recommended by CONSORT. Kindly make sure the revision follows the CONSORT format and also provide the checklist.

The manuscript has been prepared according to CONSORT guidelines for reporting randomised controlled trials. A CONSORT checklist is now provided.

Following advice from the editorial team, which relaxed the 150-word Abstract limit, we have now provided a longer CONSORT-compatible Abstract in the revised manuscript.

2) One method issue I have is that the method of allocation concealment was not mentioned in the Materials and methods section. The second is that the sample size calculation was over ambitious and has led to a small sample size and the results of this trial are now affected significantly by β error. This affects both the safety outcome of bleeding (which was more in both aspirin arms) as well as efficacy that was better in the aspirin arms. This needs to be reflected in their Discussion. With the small sample and multiple subgroup comparisons the authors need to exercise caution in the interpretation of their results. Given these concerns I feel the final conclusions should reflect no difference in primary safety or efficacy outcomes with possibility of increased harms (bleeding) as well as benefit, which needs a larger trial.

We have clarified the description of allocation concealment (subsection “Participants”).

We respectfully disagree with the reviewer on the sample size. This was a planned phase II trial which aimed to demonstrate the potential safety and efficacy of adjunctive aspirin at two different doses. As we indicate in the Introduction and the Materials and methods, there are almost no previous high-quality data upon which to construct a sample size. As acknowledged by the summary comments of the editors, and other reviewers, we have been appropriately cautious with our conclusion. We state in the Discussion that our data should not be viewed as definitive (from the perspective of clinical practice); rather that the results provide the necessary supportive data to conduct a larger phase III study. We believe our conclusions are appropriately measured. Indeed, our final sentence in the Discussion states, ‘These findings provide strong support for the conduct of a large phase 3 trial of adjunctive aspirin for TBM…’, which is as the reviewer suggests.

3) The section on sample size calculation is a bit difficult to understand. I suggest that the authors reframe this section. Here is a template that they may use to help the audience understand the trial.We calculated that with a sample size of at least ---- patients, the trial would have 80% power to detect a ----percentage-point lower risk of death among patients receiving aspirin treatment than among those receiving the standard treatment (---% vs. ---%, corresponding to a target hazard ratio of ---) at a two-sided 5% significance level.

We thank the reviewer for this suggestion. Given we conducted a 3-arm phase II trial that aimed to demonstrate both safety and potential efficacy, with little prior data on which to base the sample size calculation, such a simple description is not really possible. However, we completely accept that the current description is too long and hard to follow. Therefore, we have revised the sample size section completely (subsection “Statistical analysis”) and aimed to provide a more easily comprehensible description.

4) The main limitation is that the trial is small and underpowered even for the primary outcomes. This is a shame, as the results are promising and a slightly larger study might have shown more convincing results. The section describing the sample size is hard to follow, and would benefit from text describing more clearly what differences the study was designed to show, and to expand more on the 'clinical and feasibility considerations' that limited the sample size. Given the small sample size, and the many hypothesis tests conducted, and the resulting multiplicity considerations, the authors need to be careful not to over-state the results. Language needs to be more consistent as to what is considered evidence of a difference and what is just an observed statistically non-significant difference. There are places in the text where a difference is declared even though the p-value is greater than the common (nominal) cut-off of 0.05, e.g. subsection “Primary outcomes”, third paragraph (p=0.18) and subsection “Secondary outcomes and adverse events”, last paragraph (p=0.11). The sample size calculations talk about 'mild evidence' with a 10% one-sided significance level. If this is the threshold for evidence that the authors are using, this should be clearly stated in the Materials and methods.

We suggest that the trial can only be called underpowered in light of the results and therefore this criticism is unreasonable. We powered the trial with very limited available data to indicate likely safety or effect size. Plausibly, both the incidence of serious gastro-intestinal bleeding and new brain infarction in those who received aspirin could have been respectively higher and lower than actually occurred. But the objective of the trial was to provide much better estimates of both outcomes (and by implication, safety and efficacy), together with an analysis aspirin’s possible biological mechanism, such that the results would provide strong support for a larger and accurately powered phase 3 trial. We believe we have achieved this objective.

As discussed above, we have clarified the section on sample size.

With respect to calling ‘statistical significance’, we have made sure the text is clear that we use the nominal P<0.05 as the definition of significance. We have revised the Materials and methods and Results section accordingly.

Although pre-planned, there were 5 different sub-group analyses, and the authors found one sub-group that showed a significant difference, albeit p=0.06. The authors should comment on the issues regarding multiplicity that arise and how this impacts on the interpretation of this results. There is a clear biological rationale for this particular sub-group so the results are compelling nevertheless.

Through the text we consistently recommend caution must be applied when interpreting the results of the sub-group analysis (for example, see Discussion, fourth paragraph). However, we make it clear that the sub-groups were pre-planned and were not the result of an undisclosed trawl through multiple exploratory analyses. In addition, as acknowledged by the reviewer, each of the planned sub-groups had clear biological/clinical rationale and are therefore both justifiable and (as the reviewer agrees) ‘compelling’.